# The Influence of Danish Cancer Patient Pathways on Survival in Deep-Seated, High-Grade Soft-Tissue Sarcomas in the Extremities and Trunk Wall: A Retrospective Observational Study

**DOI:** 10.3390/cancers16234077

**Published:** 2024-12-05

**Authors:** Andrea Thorn, Kristoffer Michael Seem, Maj-Lis Talman, Bodil E. Engelmann, Michala Skovlund Sørensen, Ninna Aggerholm-Pedersen, Thomas Baad-Hansen, Michael Mørk Petersen

**Affiliations:** 1Department of Orthopedic Surgery, Rigshospitalet, University of Copenhagen, Blegdamsvej 9, 2100 Copenhagen, Denmark; kristoffer.michael.seem@regionh.dk (K.M.S.); michala.skovlund.soerensen@regionh.dk (M.S.S.); michael.moerk.petersen@regionh.dk (M.M.P.); 2Department of Pathology, Rigshospitalet, University of Copenhagen, Blegdamsvej 9, 2100 Copenhagen, Denmark; maj-lis.moeller.talman@regionh.dk; 3Department of Oncology, Herlev Hospital, University of Copenhagen, Borgmester Ib Juuls Vej 1, 2730 Herlev, Denmark; bodil.elisabeth.engelmann@regionh.dk; 4Department of Oncology, Aarhus University Hospital, Palle Juul-Jensen Blvd, 8200 Aarhus, Denmark; ninnpede@rm.dk; 5Department of Orthopedic Surgery, Tumor Section, Aarhus University Hospital, Palle Juul-Jensen Blvd, 8200 Aarhus, Denmark; thombaad@rm.dk

**Keywords:** sarcoma, high-malignant, survival, Cancer Patient Pathways

## Abstract

Soft-tissue sarcomas (STSs) are rare cancers that are difficult to diagnose and treat due to their variability. In 2009, Denmark introduced the Cancer Patient Pathways for sarcomas (CPPs) to improve the survival of sarcoma patients by accelerating the diagnosis and treatment processes. This study examined whether the CPPs improved survival in patients with high-grade STSs. Our findings show that survival has improved since the CPPs were introduced and treatment delays have been reduced. This research highlights the importance of streamlined cancer care in improving patient outcomes.

## 1. Introduction

Soft-tissue sarcomas (STSs) are rare and heterogeneous cancers that can develop in the body’s connective tissues, potentially affecting any part of the body. This variability in presentation often makes STSs difficult to diagnose [1,2]. The incidence of STSs in Denmark is approximately 250 per year, including both low- and high-grade malignant tumors [3,4]. In response to reports indicating that cancer patients in Denmark had poorer survival rates compared to other Western countries, the Danish Cancer Patient Pathways (CPPs) were introduced in 2008/2009 [5]. The Danish CPPs for sarcomas were implemented nationally in 2009.

The sarcoma CPPs include guidelines for general practitioners (GPs), outlining “warning signs” based on predefined clinical criteria to raise suspicion and initiate referral to the local orthopedic department as well as labeling cancer as an acute disease where all diagnostics and treatment should follow an acute fast track. The warning signs for STSs include a tumor size of > 5 cm, a fast-growing tumor, a tumor on or under the muscle fascia, and, for bone sarcomas, a palpable tumor in a bone or deep, persistent bone pain (without any obvious explanation). The local orthopedic department will then function as a filter that examines the warning signs and initiates further clinical examination and MR imaging that can then provide justified suspicion, resulting in a referral to one of two national Danish sarcoma centers. The date of a confirmed referral to a sarcoma center marks the beginning of the CPP timeframe. The timeframe includes predefined clinical timepoints and the recommended maximum time intervals between them. Each interval’s maximum allowed time is tailored to the type of sarcoma (STS or bone sarcoma) and its treatment plan and ends with the start of treatment [6].

The primary goal of the CPPs is to work as a fast-track system that reduces overall waiting times for cancer patients, thereby facilitating earlier diagnosis and treatment, which are anticipated to improve survival rates and patient satisfaction [6,7,8]. A study by Dyrup et al. reported significantly shorter diagnostic intervals and smaller tumor sizes at one of the national sarcoma centers in Denmark when comparing data from two years before and two years after the CPP implementation [9]. However, long-term studies evaluating the survival impact of the CPPs on sarcoma patients are lacking, and no research has yet examined the overall survival benefits following their introduction.

In this study, we aim to investigate survival outcomes in patients with deep-seated, high-grade STSs, comparing data from eight years before and eight years after the implementation of the CPPs in Denmark. We hypothesize that overall survival has improved for this patient group following the CPPs’ introduction.

## 2. Materials and Methods

### 2.1. Data Sources

This retrospective cohort study is a national study based on data from Denmark’s two sarcoma centers. The patient data for the 2000–2008 cohorts were collected separately from the two sarcoma centers. The Aarhus Sarcoma Center data was obtained via the Aarhus Sarcoma Registry (ASR), a locally based registry that has registered patient data prospectively since 1993. The registry includes information about diagnosis, treatment, tumor and patient characteristics, and follow-up, including local recurrence and death [10]. Patient data from the Rigshospitalet Sarcoma Center for the 2000–2008 cohort were identified by searching the local pathology register for STS diagnosis between 2000 and 2008. The local pathology register contains information on histological diagnosis, tumor anatomical location, size, grade, tumor depth, and where each biopsy was performed. Additional patient data were extracted from the identified patients’ medical records and the Danish CPR Registry (CRS). The CRS contains and continually updates information on migration and vital statuses for all Danish citizens [11].

Patient data from the 2010–2018 cohort were collected from the Danish Sarcoma Registry (DSR). This national sarcoma registry has collected information on Danish sarcoma patients since 2009. The DSR includes data on patient and tumor characteristics, diagnostic details, treatment specifics, local and distant recurrences, comorbidities, and mortality [3].

### 2.2. Patients

A total of 2705 patients were diagnosed with STSs in the extremities or trunk wall from 2000 to 2018. The patients included in this study were adults (>18 years) with biopsy-confirmed diagnosis of high-grade (grade 2 + 3, defined by the Trojani grading system [12]) STSs located under or going through the muscle fascia in the extremities or trunk wall. This study included patients with primary surgery at sarcoma centers, patients with unplanned excisions (whoops operations) outside sarcoma centers, and patients who did not receive surgery but only oncological treatment. We excluded the patients diagnosed with STSs in 2009 due to the differences in start-up time with the implementation of the CCPs at the two sarcoma centers. When each diagnosis was confirmed, the patient was removed from the cohort if the first contact date was in 2009. Grade 1 and borderline tumors were also excluded. In examining the remaining patient group, 18 extraskeletal bone sarcomas were found (chondrosarcoma *n* = 2, Ewing sarcoma *n* = 5, osteosarcoma *n* = 11). These patients are often treated differently and have disease trajectories that are different from other STS patients and, therefore, were excluded from this material. The 712 remaining patients were grouped into two cohorts: 309 diagnosed in 2000–2008, defined as the Pre-Cancer Patient Pathway (pre-CPP) cohort, and 403 in 2010–2018, defined as the Post-Cancer Patient Pathway (post-CPP) cohort (Figure 1).

### 2.3. Variables

Tumor size was defined as the largest diameter and was determined differently depending on the treatment. All tumor sizes from patients who had received surgery were primarily determined from the pathology report of the primary resection as the largest diameter. MRI descriptions were used if a pathology report did not document the size or the patient had not undergone surgery. The primary orthopedic surgeon’s preoperative evaluation was utilized if the MRI did not include measurements. The date of diagnosis was defined as the date of a confirmed pathological diagnosis of sarcoma from the pathology rapport. This could be from a needle biopsy, an open biopsy, or an excision biopsy. The onset of metastatic disease was defined as the date of a confirmed metastatic biopsy if one had been acquired, the time the diagnosis was made, or from the CT scan or X-ray of the lungs if a biopsy was not carried out. Time to local recurrence was measured from the time of surgery at one of the sarcoma centers until the date of the pathology report that confirmed local recurrence.

Overall survival time was measured from the biopsy date until death from any cause, emigration, or end of follow-up time (1 February 2024).

The primary endpoint was the difference in 5-year overall survival, and the secondary endpoints were the diagnostic interval days, size, local recurrence, and metastatic disease at the time of diagnosis.

### 2.4. The Danish Cancer Patient Pathway Timeframe

The timepoints and time intervals defined in the CPPs are shown in Figure 2. Timepoint A is when a referral from an orthopedic department with a justified suspicion, including a correct MR projection, is accepted by one of two sarcoma centers at a multidisciplinary meeting (MDT). When the MDT approves the justified suspicion, the CPPs are initiated. Timepoint B is the date of the first appointment at a sarcoma center and, in some cases, also the biopsy date. Timepoint C is the date of diagnosis, and Timepoint D is the first treatment date, either for surgery or oncological treatment. In this study, we only had timepoints C–D, also called time to treatment (TT). TT is the time from the approved pathologist diagnosis until the first treatment is initiated, and the maximum timeframes for this are, according to the CPPs, 14 calendar days for surgery, 15 calendar days for radiotherapy, and 11 calendar days for chemotherapy.

### 2.5. Statistical Analysis

Descriptive statistics were used to summarize the patient baseline characteristics. Continuous variables were reported as means with ranges or medians with interquartile ranges (IQRs) or ranges, depending on the data distribution. Categorical variables were summarized as counts and percentages. The Wilcoxon rank sum test was employed to compare the continuous variables between the groups, and the Chi-square test (χ^2^) was used for the categorical variables. The five-year survival probabilities were estimated using the Kaplan–Meier method, with the differences between the two groups assessed by the log-rank test. The local recurrence rates were calculated through competing risk analysis using the Aalen–Johansen estimator, considering death as a competing risk. Gray’s test was used to compare the differences in the local recurrence rates between the groups.

It was a pre-analysis, protocolized decision to dichotomize the tumor sizes into ≥5 cm and ≥15 cm, aiming to avoid overestimating clinical effects in external populations. The two patient groups were 2000–2008 (pre-CPP) and 2010–2018 (post-CPP). We considered *p*-values of < 0.05 as statistically significant and reported confidence intervals as 95% (95% CIs). Analyses were performed using R, version 4.3.0 (R Development Core Team, Vienna, Austria, 2020).

## 3. Results

### 3.1. Patients and Tumor Characteristics

A total of 712 patients with deep-seated, high-malignant STSs were included in this study, 309 from 2000 to 2008 and 403 from 2010 to 2018, and not-otherwise-specified sarcoma (NOS) diagnosis was the most common histological diagnosis in both cohorts (Table 1).

Both cohorts showed no statistically significant differences regarding age, tumor size, and location, but there was a difference regarding sex (*p* = 0.045), with more males (57%) in the pre-CPP cohort than in the post-CPP cohort (49%) (Table 2).

When looking at patient treatment (Table 3), a higher number of patients in the pre-CPP cohort underwent unplanned excisions (whoops operations) (10% vs. 4%, *p* < 0.001), and a tendency toward fewer amputations in the post-CPP cohort (8% vs. 12%, *p* = 0.07) was found.

Regarding the oncological treatment, more patients received chemotherapy in the post-CPP cohort (41% vs. 28%, *p* = 0.02), and a higher number of patients in the post-CPP cohort received radiotherapy, especially visible within the first three months of surgery (57% vs. 47%, *p* < 0.001).

### 3.2. Overall Survival

At the end of the study period, 477 (67%) of the patient population had died: 236 (76%) in the pre-CPP cohort and 241 (60%) in the post-CPP cohort. All patients who did not receive surgery died before five years had passed from their diagnosis date. The mean follow-up was 11 years (1–24 years). Two patients emigrated, one after one year and another after 3.6 years, but all other patients had a minimum follow-up of 5 years or until death.

The overall five-year survival rate for all patients treated between 2000 and 2018 was 48% (95 CI: 45–52), and focusing on the two groups, the probability of survival at five years for the pre-CCP cohort was 43% (95 CI: 38–49), and it was 52% (95 CI: 47–57) for the post-CPP cohort. A clear tendency toward an improved overall survival rate in the in the post-CPP cohort was found (*p* = 0.05) (Figure 3).

### 3.3. Local Recurrence

A total of 133 (19%) of the total patient population developed local recurrence: 57 (18%) in the pre-CPP cohort and 76 (19%) in the post-CPP cohort. The overall cumulative incidence of local recurrence at five years was 15% (95% CI: 11–19) for the pre-CPP cohort and 17% (95%: 14–21) for the post-CPP cohort, and no difference in local recurrence rates between the two groups could be found (*p* = 0.52) (Figure 4).

### 3.4. Diagnostic Time Intervals

Timeframes C–D were lower in the post-CPP cohort. The pre-CPP cohort had a median of 18 days (IQR: 12–30) and the post-CPP cohort had one of 15 days (IQR: 9.25–21) (*p* < 0.001).

## 4. Discussion

We evaluated the OS of deep-seated, high-malignant STSs in the extremities and trunk wall before and after introducing fast-track CPPs. We found that the post-CPP cohort had a clear tendency toward a higher overall 5-year survival rate than the pre-CPP cohort (52% vs. 43%). Most studies report a 5-year overall survival for STSs of between 50% and 72%, depending on subtype, histological grade, depth, local recurrence, tumor size, and patient age. Deep-seated, high-grade STSs generally have worse overall survival than low-grade, superficial STSs [13,14,15,16,17]. The patients in our population were all high-grade and subfascial, and 4% of them had metastatic disease at presentation. When patients present with metastatic disease, their 5-year survival decreases substantially to 6–24% [18,19]. All these factors likely contributed to the level of the 5-year OS seen in the present study. In a registry study with the same patient group in Denmark, metastatic patients were excluded, and the 5-year overall survival for non-metastatic subfascial high-grade patients in Denmark from 2000 to 2016 was 60%, aligning more with the general research in the area [20]. Our study’s cumulative five-year local recurrence rate was 16%, similar to other studies that included highly malignant tumors such as ours [21,22].

When comparing the two cohorts, we found a significant difference in the number of patients that received oncological treatment. The proportion of patients receiving radiotherapy within the first three months after surgery increased from 47% to 57%, and the proportion of patients receiving chemotherapy increased from 28% in the pre-CPP cohort to 41% in the post-CPP cohort. The specific oncological treatments, e.g., radiotherapy dose, beam energy, fraction size, and the type of chemotherapy regime, were not readily available for the present study. Generally, by introducing the CPPs in our country, there was a greater emphasis on the timely administration of radiation therapy and selected high-risk patients received more chemotherapy.

Additionally, the two centers have more aggressive strategies toward oligometastatic disease and palliative treatment for patients with metastatic STSs that evolved during the time of this study. Furthermore, a widened range of treatment options have been available for the post-CPP cohort [23,24].

STS is a very heterogeneous type of cancer with over 70 different subtypes, and more are being added with the integration of morphologic, immunohistochemical, and molecular characteristics to the diagnostic process [25]. Around 2002, the histological diagnosis of malignant fibrous histiocytoma (MFH) was beginning to be replaced with the diagnosis of undifferentiated pleomorphic sarcoma (UPS) in most cases [26]. In our data, this is seen as the pre-CPP cohort having no UPS diagnosis. The most common type is sarcoma NOS, with 64 patients in the pre-CPP cohort and 81 in the post-CPP cohort. Even with a large overall study population, these numbers are still too small to make any subgroup analysis. Hopefully, more studies on specific subtypes will emerge with the growing number of registers worldwide.

The rarity of STSs, coupled with a general lack of awareness in the public and within the healthcare system, can result in long intervals of time to diagnosis or misdiagnosis. This delay can result in larger tumor sizes and a higher proportion of patients with metastatic disease at the time of diagnosis. One of the aims of the CPPs was to expedite referral to sarcoma centers by easily identifiable warning signs that would qualify patients for early referral, hopefully leading to earlier diagnosis, more patients with smaller tumors, and fewer patients with localized disease. However, our findings revealed no significant difference in the proportion of patients presenting with metastatic disease before and after the implementation of the CPPs, and similarly, tumor size showed only a very weak tendency toward more tumors larger than 5 cm and larger than 15 cm in the pre-CPP cohort. These findings contrast with those of Dyrup et al., who reported a significant reduction in tumor size at a Danish sarcoma center two years after the introduction of the CPPs [9]. Our study showed that this effect was diminished when data were examined in a larger sample and over a longer time span before and after CPP implementation. It is well-established that tumor size plays a critical role in predicting mortality, with larger tumors at diagnosis being associated with poorer survival outcomes [16]. This suggests that the better five-year survival found in this study is less attributable to tumor size. The initial focus and information campaign when the CPPs were introduced in 2009 could also explain why tumor size reductions were initially observed but have since plateaued. Dyrup et al. also examined the presence of alarm symptoms two years before and two years after the CPPs [27]. They showed that approximately one-third of the patients presented without alarm symptoms and had STSs that were found accidentally [27]. MRI has become more available in recent years, mainly due to reduced costs and improved accessibility. MRI is a highly sensitive tool for detecting soft-tissue abnormalities, and its frequent use could contribute to accidentally detecting sarcomas earlier, especially in patients without overt alarm symptoms. Still, research shows that one or more alarm symptoms increase the risk of STS malignancy, so a continuous information campaign on sarcoma and potentially further research to expand the warning signs are warranted [28].

Another aspect of the CPPs was to establish clear criteria for justified suspicion. This could potentially reduce unplanned sarcoma excisions (whoops procedures) performed at non-specialized facilities and encourage referral to sarcoma centers. Our findings show a significant reduction in whoops procedures in the post-CPP group (10% vs. 4%). Whoops procedures carry a 24–91% risk of intralesional margins, which are associated with a high likelihood of local recurrence, metastasis, and poorer overall survival [29]. Although most of these patients will undergo re-resection, which may mitigate some of the risks to those similar to a planned excision, studies have indicated that second surgeries are often more extensive than initial ones; are linked to a higher incidence of complications, including amputation; and should be avoided [17,29,30].

Our study demonstrated a significant reduction in the median time-to-treatment interval (timeframe C–D) of three days (reduced from 18 days to 15 days) following the implementation of the CPPs. This reduction suggests that the CPPs have successfully sped up the TT intervals for sarcoma patients. Time-to-treatment (TT) intervals exceeding 30 days have been associated with decreased survival in patients with STSs [31]. However, since the pre- and post-CPP time intervals in our study were both under 30 days, it is unclear if this reduction directly impacted the OS. Timepoints A and B were unavailable for many patients in our study, preventing a direct comparison of the total time interval from A to D before and after CPP implementation. Dyrup et al. also reported a significant reduction in the total time interval (A–D) for STS when comparing data from two years before and two years after CPP implementation [9]. Although we could not assess the full A–D interval, our study’s observed decreases in the TT suggest a potential reduction in the overall timeframe for sarcoma patients.

Despite the reductions in the TT intervals in our cohort, it remains unclear whether these shortened intervals translated into improved survival outcomes for our STS patients. Given sarcomas’ biology and clinical behavior heterogeneity, the relationship between time and treatment is hard to determine.

Several factors during the study period could have influenced patient survival outcomes. Like many other countries, Denmark has experienced a centralization of sarcoma treatment. Multiple studies, including those focused on STSs, have demonstrated improved survival rates for patients treated at specialized sarcoma centers. These studies emphasize the significant survival benefits of receiving care in specialized high-volume institutions [32,33]. Furthermore, life expectancy also increased during this period. In 2000, the life expectancy for women was 79.2 years, and it was 74.5 years for men. By 2018, this had risen to 82.9 years for women and 79 years for men [34]. These improvements in life expectancy align with the broader trends of enhanced centralization and cancer survival, underscoring the positive impact of national health strategies over time.

A limitation of this study is its potential lack of reproducibility in countries with different healthcare systems. Denmark’s healthcare system is publicly funded and highly centralized, with two specialized sarcoma centers, ensuring uniformity in treatment protocols and follow-up. However, similar studies may encounter difficulties in replicating our findings in more fragmented healthcare systems that rely on private healthcare providers or have less centralized sarcoma care. Differences in access to care, variations in treatment protocols, and incomplete data collection may limit the external reproducibility of our results in other healthcare settings. The strengths of our study are the large number of high-malignant STS patients, few missing data, and a high number of variables collected. However, this is a retrospective registry study, and our analysis is, therefore, inherently reliant on the accuracy and completeness of data entry by multiple doctors throughout multiple specialties, which directly influences the reliability and validity of these findings. In the case of our national sarcoma registry, most data are entered manually, increasing the potential for information bias. This can arise from errors in data recording, incomplete records, or inconsistencies in data entry practices across different centers. To minimize the risk of information bias, the same person conducted regular cross-checks and validation against primary sources where possible to ensure the integrity of the data used in the analysis. However, despite these efforts, the possibility of residual bias could not be entirely eliminated, and this should be considered when interpreting this study’s results.

## 5. Conclusions

This study showed a clear tendency toward better overall 5-year survival after the introduction of the CPPs (52% vs. 43%), suggesting an effect of the Danish CPPs on sarcomas. We found only a very weak tendency toward larger tumors in the pre-CPP cohort and no difference regarding the percentage of patients that had distant metastases at diagnosis between the cohorts. However, in the post-CPP cohort, the percentage of whoops operations was decreased, the use of oncological services increased, and the time-to-treatment interval was reduced. This study highlights the importance of efficient referral pathways in improving cancer outcomes, but it cannot exclude that other factors could also have contributed to improved survival.

## Figures and Tables

**Figure 1 cancers-16-04077-f001:**
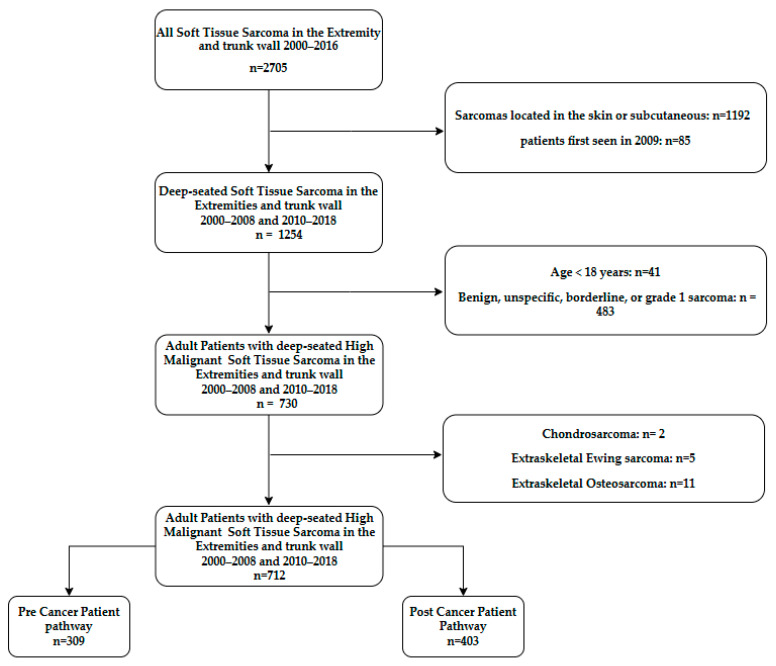
Flowchart.

**Figure 2 cancers-16-04077-f002:**
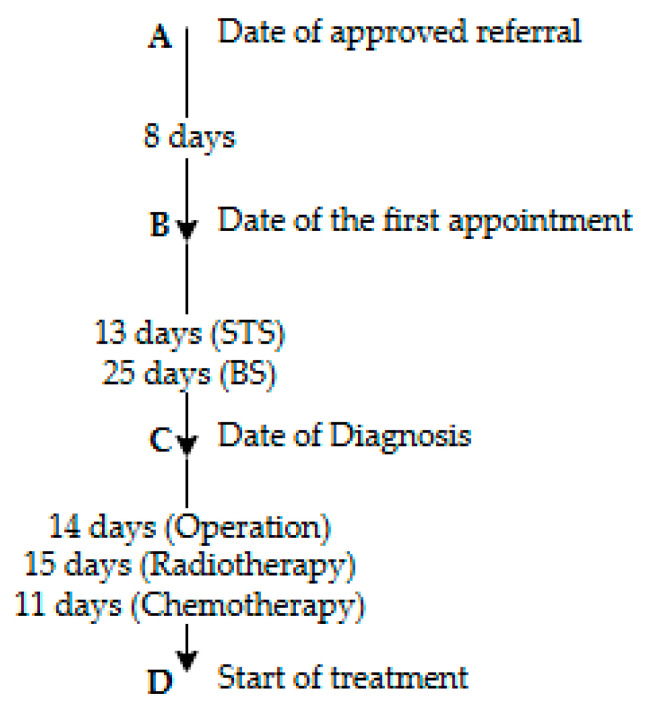
The total timeframe for the Danish patient pathways for sarcoma. STS = soft-tissue sarcoma, BS = bone sarcoma.

**Figure 3 cancers-16-04077-f003:**
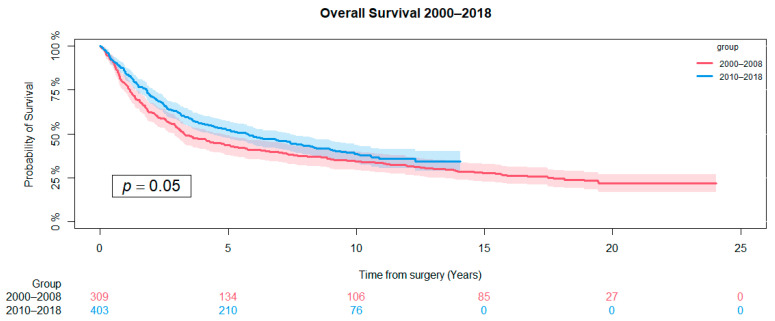
Kaplan–Meier graph showing the difference in survival in the two groups. The difference was evaluated using the log-rank test.

**Figure 4 cancers-16-04077-f004:**
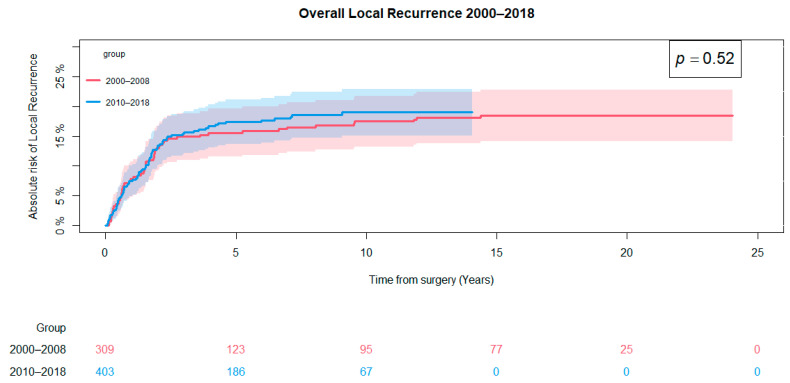
Overall cumulative incident of local recurrence. The difference between the two cohorts was evaluated using Gray’s test.

**Table 1 cancers-16-04077-t001:** Most common histological diagnoses. NOS (not otherwise specified).

Treatment group	Overall	2000–2008	2010–2018
	(*n* = 712 ^1^)	(*n* = 309 ^1^)	(*n* = 403 ^1^)
**Histological diagnosis**			
Sarcoma NOS	145 (20%)	64 (21%)	81 (20%)
Undifferentiated pleomorphic sarcoma	70 (10%)	0 (0%)	70 (17%)
Leiomyosarcoma	68 (10%)	20 (9%)	39 (10%)
Myxoid liposarcoma	62 (9%)	27 (9%)	35 (9%)
Malignant fibrous histiocytoma	59 (8%)	54 (17%)	5 (1%)
Myxofibrosarcoma	57 (8%)	4 (2%)	53 (13%)
Other	251 (35%)	131 (42%)	120 (30%)

^1^ Frequency (%).

**Table 2 cancers-16-04077-t002:** Overview of patients’ characteristics.

Treatment Group	Overall	2000–2008	2010–2018 *p*-Value ^3^	
Overall	(*n* = 712 ^1^)	(*n* = 309 ^1^)	(*n* = 403 ^1^)	
Sex				0.045
Female	337 (47%)	133 (43%)	204 (51%)	
Male	375 (52%)	176 (57%)	199 (49%)	
Age (Years ^2^)	62 (19–96)	60 (19–91)	63 (19–96)	0.052
Location				0.7
Lower Extremity	499 (70%)	220 (71%)	279 (69%)	
Truncal	70 (10%)	27 (9%)	43 (11%)	
Upper Extremity	143 (20%)	62 (20%)	81 (20%)	
Tumor Size (cm)	10.7 (1–40)	10.8 (1–30)	10.6 (1–40)	0.4
Tumor Size				0.2
<5 cm	86 (12%)	32 (10%)	54 (13%)	
>=5 cm	626 (88%)	277 (90%)	349 (87%)	
Tumor Size				0.3
<15 cm	541 (76%)	229 (74%)	312 (77%)	
>=15 cm	171 (24%)	80 (26%)	91 (23%)	

^1^ Frequency (%); ^2^ age: mean (range); ^3^ Pearson’s Chi-square test, Wilcoxon rank-sum test.

**Table 3 cancers-16-04077-t003:** Overview of patients’ treatment and baseline conditions.

Treatment Group	Overall	2000–2008	2010–2018 *p*-Value ^2^	
Overall	(*n* = 712 ^1^)	(*n* = 309 ^1^)	(*n* = 403 ^1^)	
Surgical Margin				0.7
Intralesional	76 (11%)	28 (9%)	48 (12%)	
Marginal Wide No Surgery	272 (38%) 288 (40%) 76 (11%)	120 (39%) 128 (41%) 33 (11%)	152 (38%) 160 (40%) 43 (11%)	
Amputation (Only Extremities)	64 (10%)	35 (12%)	29 (8%)	0.070
Whoops Procedures	47 (7%)	32 (10%)	15 (4%)	<0.001
Local Recurrence	133 (19%)	57 (18%)	76 (19%)	>0.9
Chemotherapy				0.002
<3 months of Surgery	22 (3%)	5 (2%)	17 (4%)	
>3 months of Surgery Before Surgery Chemo, No Surgery No Chemotherapy	161 (23%) 30 (3%) 40 (6%) 459 (64%)	61 (20%) 9 (3%) 11 (4%) 223 (72%)	100 (25%) 21 (5%) 29 (7%) 236 (59%)	
Radiotherapy				<0.001
<3 months of Surgery	374 (52%)	146 (47%)	228 (57%)	
>3 months of Surgery	79 (11%)	50 (16%)	29 (7%)	
Before Surgery Radiation, No Surgery No Radiation	8 (1%) 39 (6%) 212 (30%)	5 (2%) 11 (4%) 97 (31%)	3 (1%) 28 (7%) 115 (29%)	
Metastasis				0.9
No Metastatic Disease	349 (49%)	150 (49%)	199 (49%)	
At Diagnosis <3 months of Diagnosis >3 months of Diagnosis	31 (4%) 63 (9%) 269 (38%)	15 (5%) 27 (9%) 117 (38%)	16 (4%) 36 (9%) 152 (38%)	

^1^ Frequency (%), ^2^ Pearson’s Chi-square test; Wilcoxon rank-sum test.

## Data Availability

The data presented in this study are restricted due to patient confidentiality and are, therefore, not publicly available. However, they can be made available to the corresponding author on reasonable request.

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
