# Peer review of "The Influence of Danish Cancer Patient Pathways on Survival in Deep-Seated, High-Grade Soft-Tissue Sarcomas in the Extremities and Trunk Wall: A Retrospective Observational Study"

_cancers, 2024, doi:10.3390/cancers16234077_

Round 1
Reviewer 1 Report
Comments and Suggestions for Authors
Interesting study showing again the importance of centralized care in specific pathologies such as soft tissue sarcoma.
The limits of the study are clearly defined.
I would refine the quality of the charts because they appear pixelized.
Question regarding the statistical M&M: lines 160-161. Authors states they compared the results between the two centers. There are no results showed for this and it is not discussed afterwards. What was the goal, was it done? Maybe show the results and discussed them. There should not be any difference.
Regarding the use of radiotherapy, was there an increase in use of preop RT following the introduction of CPP?
Any idea as to why the local recurrence rate increases slightly in post CPP? Even if not statistically significant.
Author Response
We sincerely thank the reviewers for their thoughtful and constructive feedback. Each comment has been carefully considered and addressed, and the manuscript has been revised accordingly.
Interesting study showing again the importance of centralized care in specific pathologies such as soft tissue sarcoma.
The limits of the study are clearly defined.
I would refine the quality of the charts because they appear pixelized.
Thank you for your comment. Both Figures 1 and 2 have been revised to improve their quality and resolution.
Question regarding the statistical M&M: lines 160-161. Authors states they compared the results between the two centers. There are no results showed for this and it is not discussed afterwards. What was the goal, was it done? Maybe show the results and discussed them. There should not be any difference.
This was a typographical error, which has been corrected to clarify that comparisons were made between groups, not centers.
Regarding the use of radiotherapy, was there an increase in use of preop RT following the introduction of CPP?
No, as shown in Table 3, preoperative radiotherapy was used in 5 patients (2%) in the pre-CPP group and three patients (1%) in the post-CPP group.
Any idea as to why the local recurrence rate increases slightly in post CPP? Even if not statistically significant.
We would not call this an increase: percentage of local recurrence 18% (18.4%) versus 19% (18.9%) (p>0.9), and risk of local recurrence rate compared by Gray´s test (p=0.52). It is a very week tendency which might be explained by a higher percentage of patients that ended up with tumor removal by intralesional margin and fewer amputations in the post-CPP group. Since this was such a week tendency, we have not commented on this in the Discussion, but if it is in the judgement of the editor, we can of course do it.
Reviewer 2 Report
Comments and Suggestions for Authors
This is a great research paper which has the potential to change practice in other countries - I really enjoyed reading this. Congratulations!
There were some very minor grammatical errors which should have been corrected prior to publication.
Change to: Soft tissue sarcomas (STS) are rare and heterogeneous cancers – line 46
Change to: potentially affecting any part of the body – line 47
This sentence is not understandable: “a tumor on or profound for the deep fascia” and needs to be edited – line 58
This sentence should be changed – please clearly define Whoops as an unplanned excision, rather than “operated on outside a sarcoma center before being referred to one (Whoops operations)” line 106
“a Whoops operation” would change this to unplanned excision line 185
Definition given “unplanned sarcoma excisions (Whoops procedures)” but Whoops should be clearly defined at the beginning line 288
Author Response
We sincerely thank the reviewers for their thoughtful and constructive feedback. Each comment has been carefully considered and addressed, and the manuscript has been revised accordingly.
This is a great research paper which has the potential to change practice in other countries - I really enjoyed reading this. Congratulations!
Thank you for your kind words and encouragement.
There were some very minor grammatical errors which should have been corrected prior to publication.
Change to: Soft tissue sarcomas (STS) are rare and heterogeneous cancers – line 46
The suggested change has been made.
Change to: potentially affecting any part of the body – line 47
The suggested change has been made.
This sentence is not understandable: “a tumor on or profound for the deep fascia” and needs to be edited – line 58
The line has been revised for clarity.
This sentence should be changed – please clearly define Whoops as an unplanned excision, rather than “operated on outside a sarcoma center before being referred to one (Whoops operations)” line 106
The line has been revised to define "Whoops operations" as unplanned excisions.
“a Whoops operation” would change this to unplanned excision line 185
The revision has been made.
Definition given “unplanned sarcoma excisions (Whoops procedures)” but Whoops should be clearly defined at the beginning line 288
The suggested correction has been incorporated.
Reviewer 3 Report
Comments and Suggestions for Authors
In this manuscript, the authors describe a retrospective observational study regarding "The Influence of Danish Cancer Patient Pathways on Survival in Deep-Seated High-Grade Soft Tissue Sarcomas in the Extremities and Trunk Wall". While this manuscript is generally well written, it would be helpful if the authors address the minor concern below:
1) Figures 1 and 2 are very blurry and their fonts are inconsistent with the entire manuscript. The authors should address that.
2) As the authors rightly pointed out, while this study has a very narrow scope with special reference to a specific country, data from this study would not be reproducible due to a wide variation in the health delivery systems globally. Regardless, this study provides an alternative model with modest positive patient outcomes worth taking note of.
In summary, this study could benefit its target readers and could potentially provide a different perspective regarding patient care in cases of deep-seated high-grade soft tissue sarcomas, as well as cancer care in general.
Author Response
We sincerely thank the reviewers for their thoughtful and constructive feedback. Each comment has been carefully considered and addressed, and the manuscript has been revised accordingly.
In this manuscript, the authors describe a retrospective observational study regarding "The Influence of Danish Cancer Patient Pathways on Survival in Deep-Seated High-Grade Soft Tissue Sarcomas in the Extremities and Trunk Wall". While this manuscript is generally well written, it would be helpful if the authors address the minor concern below:
- Figures 1 and 2 are very blurry and their fonts are inconsistent with the entire manuscript. The authors should address that.
Figures 1 and 2 have been revised to ensure improved resolution and consistent formatting.
- As the authors rightly pointed out, while this study has a very narrow scope with special reference to a specific country, data from this study would not be reproducible due to a wide variation in the health delivery systems globally. Regardless, this study provides an alternative model with modest positive patient outcomes worth taking note of.
Thank you for your valuable observation.
In summary, this study could benefit its target readers and could potentially provide a different perspective regarding patient care in cases of deep-seated high-grade soft tissue sarcomas, as well as cancer care in general.
We appreciate your positive feedback.